# Solar-Light-Driven Efficient ZnO–Single-Walled Carbon Nanotube Photocatalyst for the Degradation of a Persistent Water Pollutant Organic Dye

**Kamal Prasad Sapkota** [1,2]**, Insup Lee** [1]**, Md. Abu Hanif** [1] **, Md. Akherul Islam** [1] **and Jae Ryang Hahn** [1,3,*]

[1]   Department of Chemistry and Bioactive Material Sciences, Research Institute of Physics and Chemistry, Chonbuk National University, Jeonju 54896, Korea; mychemistry2037@gmail.com (K.P.S.); insup@jbnu.ac.kr (I.L.); hanif4572@gmail.com (M.A.H.); akherulraju@gmail.com (M.A.I.)

[2]   Department of Chemistry, Amrit Campus, Tribhuvan University, Kathmandu 44618, Nepal

[3]   Textile Engineering, Chemistry and Science, North Carolina State University, 2401 Research Dr., Raleigh, NC 27695-8301, USA

*   Correspondence: jhahn@ncsu.edu or jrhahn@jbnu.ac.kr

**Abstract:**   An efficient photocatalyst, ZnO–single-walled carbon nanotube (ZnO–SWCNT) nanocomposite was successfully fabricated through a straightforward one-pot–two-chemical recrystallization technique followed by thermal decomposition. The photocatalytic efficiency of the prepared ZnO–SWCNT composite was investigated by assessing the degradation of a persistent water-pollutant dye (methylene blue, MB) under visible-light irradiation. We found that the synthesized photocatalyst is an effective and recyclable agent for the decomposition of an MB solution. Its photocatalytic performance was substantially better than that of pristine ZnO nanorods or pristine SWCNTs. The reusability of the photocatalyst was also examined, affirming that it could be used repeatedly for five cycles without conspicuous loss of morphology or catalytic performance.

**Keywords:** ZnO–SWCNT composites; recrystallization; visible light; photocatalyst; dye; decomposition

## 1. Introduction

Severely toxic and nonbiodegradable organic contaminants such as dyes, phenoxyanilines, phenols, and their derivatives are commonly present in industrial effluents [1]. Among these contaminants, textile dyes are increasingly problematic because of their release into water resources [2]. The residual liquid after an industrial dyeing operation contains approximately 15% dyes. However, it is often disposed of directly into nearby water bodies, creating severe environmental pollution [3]. Such pollutants exhibit genotoxicity and pose a threat to the human endocrine system, even at low concentrations [1]. Furthermore, some dyes are toxic, mutagenic, or carcinogenic, whereas others affect aquatic biota by reducing light penetration and hindering photosynthesis [3]. Hence, extensive research is underway to develop a suitable process for environmental decontamination [4]. Thus far, the most extensively exploited techniques for dye removal include chemical oxidation, coagulation, adsorption, and photocatalysis. Photocatalysis is becoming an increasingly popular choice for the degradation of pollutants, particularly for wastewater with relatively low concentrations of recalcitrant organic constituents. Photocatalysis offers numerous advantages over alternative techniques; these advantages include total mineralization, which avoids subsequent waste disposal problems; low operating costs; easy accessibility; high performance; and efficient action under ambient environmental conditions [5,6].

Zinc oxide (ZnO) is a semiconductor with an exciton binding energy (60 meV), large bandgap (3.37 eV), and outstanding stability against thermal decomposition. Nanosized ZnO materials have been

demonstrated to be very useful in a large number of technical applications, including photocatalysts, photonic crystals, photodetectors, transparent conducting oxides, biosensors, photoelectrochemical water-splitting devices, solar cells, bone tissue engineering scaffolds, and supercapacitors [7–10]. Nanosized and/or nanostructured ZnO is a suitable photocatalyst because of its nontoxicity, low cost, wide bandgap, availability of a sizeable active portion (or area) for photocatalytic reaction, large aspect ratio, and high photosensitivity [1–3]. However, there are some nuisances in the practical application of ZnO as a photocatalyst. Its narrow spectral response range and high probability for the recombination of photogenerated electron–hole pairs are two such issues [11,12].

Carbon nanotubes (CNTs) exhibit several interesting properties, including high thermal and electrical conductivities, large electron-storage capacity, and extremely high tensile strengths. They can easily form uniform composites with nanosized semiconductor metal oxides [13–15]. The composites of ZnO and CNTs have found important applications in diverse fields such as photocatalysis, photonic devices, microelectronics, optical devices, coatings, and drug and gene delivery [16–18]. Hybrid nanocomposites consisting of a layer of crystalline ZnO coating, CNTs have been extensively used as an important component in nanosensors, nanogenerators, and nanoresonators [19,20]. A ZnO–single-walled carbon nanotube (ZnO–SWCNT) nanocomposite has been reported to enhance the stability and efficiency of organic solar cells [21].

Researchers have developed several innovative approaches to enhancing the photocatalytic activity of ZnO. Several such studies have revealed that the coupling between ZnO and other semiconductors can increase the distance between photogenerated electron–hole pairs and adjust the bandgap energy [22]. Furthermore, in nanocomposites comprising a semiconductor oxide (such as ZnO) and CNTs, the CNTs act as excellent electron acceptors or electron transport materials and strongly promote the conduction of photogenerated electrons, resulting in enhanced photocatalytic activity. This phenomenon suppresses the likelihood of photogenerated electron–hole pairs recombining and improves the catalytic efficiency [17,23,24]. Han et al., for example, reported an improvement of the photocatalytic and photocorrosion-resistance activity of ZnO by compositing it with multipurpose carbon constituents such as CNTs, fullerene ($C_{60}$), graphene, and other carbon allotropes [25]. Bartfai et al. fabricated an efficient ZnO-multi-walled carbon nanotube (ZnO–MWCNT) photocatalyst and used it in the photocatalytic decomposition of acetaldehyde [26]. Chaudhary et al. synthesized ZnO-decorated MWCNTs and used them to photocatalyze the decomposition of methylene blue (MB) and to photoelectrochemically split water to generate hydrogen [27]. Hossain et al. fabricated a thermally stable solar-light-driven photocatalyst from a nanocomposite of ZnO and CNT fibers and used it to efficiently photodegrade MB [28]. Ohno et al. used a modified Ga:ZnO solid solution to photocatalytically split water [29]. Turkyilmaz et al. doped ZnO with Ni, Mn, Fe, and Ag and found that the resultant composite exhibited improved photocatalytic efficiency compared with that of pristine ZnO [30]. Trandafilovic et al. fabricated Eu-doped ZnO nanoparticles and reported that they exhibit superior photocatalytic activity toward both methyl orange and MB [31]. Zhao et al. used ZnO particles with different shapes and proportions to adorn reduced graphene oxide and found that an adequate quantity of ZnO nanorings adorning reduced graphene oxide greatly enhanced the absorption of visible light and the intensity of photoluminescent emission [32]. Improvements in the photocatalytic performance of ZnO through doping with other metals or through composite formation have been reported in several other works [33–37]. Furthermore, the repeatable applicability of the nanomaterial in photocatalysis has been demonstrated to be affected by the control of properties of ZnO nanostructures, especially size and shape [38–40].

To our knowledge, the literature contains no reports describing the design and synthesis of ZnO–SWCNT photocatalysts with remarkable electron–hole separation efficiency and enhanced photocatalytic propensity. In this paper, we report the synthesis of a ZnO–SWCNT nanocomposite via a low-cost, one-pot recrystallization technique. The photocatalytic performance of the constructed nanocomposite was explored through its use as a catalyst for the decomposition of MB under sunlight. The influence of distinct factors—specifically, the concentration of the dye solution, amount of

photocatalyst, and the exposure time—was also investigated. This report presents the photocatalytic proficiency of the prepared nanocomposite at the optimum values of these parameters. The nanocomposite consisting of SWCNTs and ZnO nanorods impedes the recombination of photoinduced electron–hole pairs and enhances the catalytic performance compared with that of ZnO or SWCNTs alone. We reused the photocatalyst five times without notable degradation in performance, although a trivial change in its morphology as a result of repeated handling was observed.

## 2. Results and Discussion

### 2.1. Morphological Characterization of the ZnO–SWCNT Nanocomposite

Figure 1 displays typical images of the ZnO–SWCNT nanocomposite, as observed via FE-SEM. Each SWCNT is surrounded by several ZnO nanorods. Some of the SWCNTs are indicated by red arrows for easy observation (see also the further analysis by HR-TEM later in this section). A random orientation of ZnO nanorods is observed around the SWCNTs. The space between successive clusters of ZnO nanorods shows the separation between CNTs. The area where the SWCNTs are closer to one another is occupied by more crowded clusters of ZnO nanorods, whereas the SWCNTs are somehow scattered in other locations. The ZnO nanorods have various lengths and diameters, with random spatial orientation. The ZnO nanorods appear to be within the nanometer size range, whereas the size of the composites appears to depend on the number of nanorods surrounding the SWCNTs.

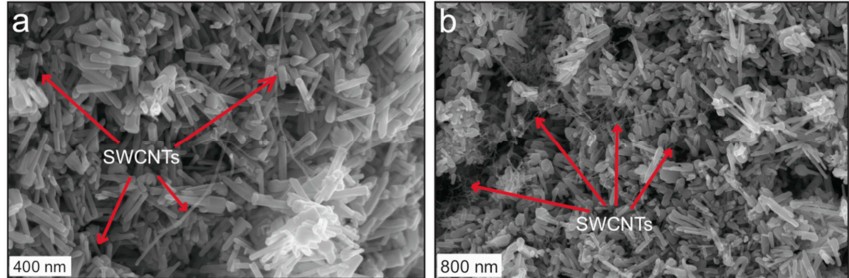

**Figure 1.** (**a**,**b**) FE-SEM images of the ZnO–single-walled carbon nanotube (SWCNT) nanocomposites. ZnO nanorods are surrounded by SWCNTs (indicated by the arrows in each image) in the ZnO–SWCNT nanocomposite.

To identify the chemical components of the ZnO–SWCNT nanocomposite, we used energy-dispersive X-ray spectroscopy (EDX) to analyze the sample during the FE-SEM observation. Figure 2b presents the EDX spectrum of the prepared sample, which shows the presence of Zn, O, and C. The EDX spectrum corresponds to the indicated rectangular area in the SEM image (Figure 2a). It is consistent with the results of the analysis based on FE-SEM images; i.e., each SWCNT is surrounded by a large number of ZnO nanorods (Figure 2c).

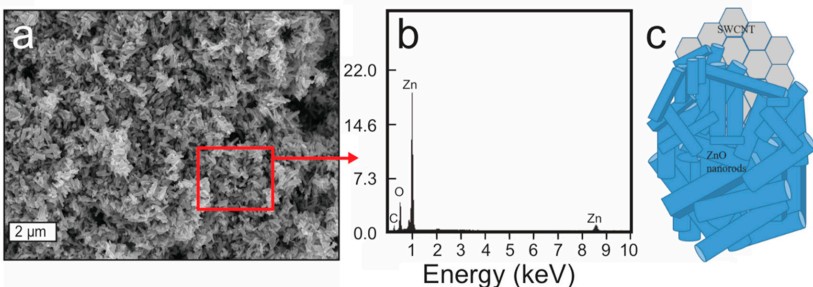

**Figure 2.** (**a**) FE-SEM micrograph of the ZnO–SWCNT nanocomposite. (**b**) An EDX spectrum corresponding to the region in the rectangle in Figure 2a. (**c**) Schematic of the hybrid nanocomposite structure in which an SWCNT is covered with ZnO nanorods.

## 2.2. Structural Characterization of the ZnO–SWCNT Nanocomposite

The structure of the ZnO–SWCNT nanocomposite was characterized by XRD and compared with the structures of the pristine ZnO and pristine SWCNTs (Figure 3). The spectrum of pristine SWCNTs (Figure 3c) shows peaks at $2\theta = 11.94°$, $25.71°$, and $43.06°$, which are assigned to the (001), (002), and (100) planes of graphitic carbon, respectively [16]. The maximum at $25.71°$ indicates the *d*-spacing of crystalline carbon, whereas the peak at $43.06°$ is associated with disordered carbon [28]. Similarly, the spectrum of pristine ZnO (Figure 3b) shows characteristic diffraction peaks at $31.84°$, $34.52°$, $36.33°$, $47.63°$, $56.71°$, $62.96°$, $68.13°$, and $69.18°$. All of these peaks correspond to the crystalline wurtzite structure of ZnO in hexagonal geometry (JCPDS No. 36-1451) [12]. The crystallite size of ZnO was calculated from XRD using Scherrer equation [$D = k \lambda/(\beta \cos\theta)$, where '$D$' is the crystallite size, '$k$' is Scherrer constant (i.e., 0.9), '$\lambda$' is the wavelength of X-ray, '$\beta$' is the full width at half maximum (FWHM) of the main peak and '$\theta$' is the diffraction angle]. The crystallite size was 33.2 nm. The reflection from (101) plane occurring at 36.27 ($2\theta$) was used for the measurement of crystallite size.

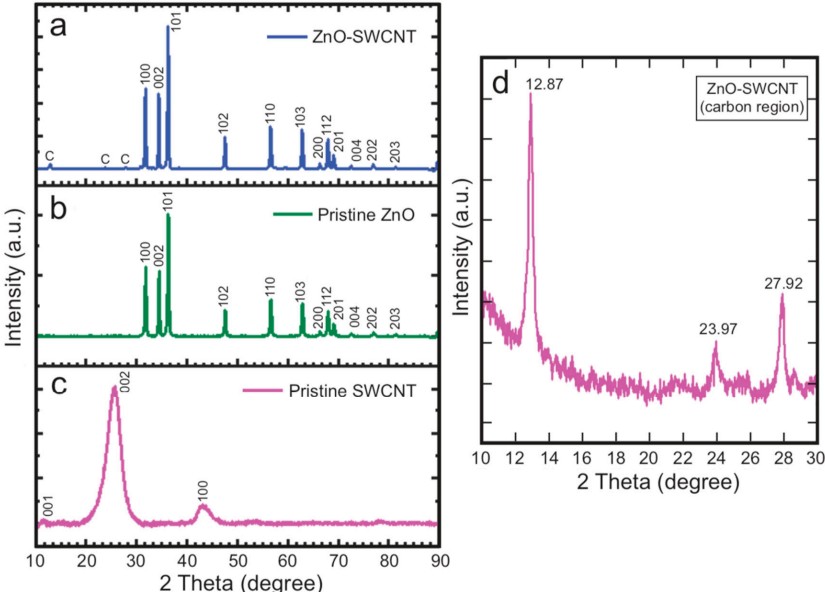

**Figure 3.** XRD patterns of (**a**) the ZnO–SWCNT nanocomposite, (**b**) the pristine ZnO, and (**c**) the pristine SWCNTs; (**d**) magnified peaks in the carbon portion of the XRD pattern of the nanocomposite.

The magnified spectrum of the ZnO–SWCNT nanocomposite (Figure 3d) shows carbon peaks at $12.87°$, $23.97°$, and $27.92°$. The additional peaks in Figure 3a are attributed to diffraction from ZnO at different angles. The peaks representing *hkl* planes (100), (002), (101), (102), (110), (103), (200), (112), (201), (004), and (202) of ZnO were detected in the XRD pattern of the ZnO–SWCNT nanocomposite. These observations suggest that the hexagonal crystalline wurtzite phase of ZnO remains unchanged in the ZnO–SWCNT composite (JCPDS-36-1451) [1].

The carbon peaks in the XRD pattern of the composite are somewhat shifted from those in the pattern of the pristine SWCNTs; i.e., the peaks at $25.71°$ and $43.06°$ in the XRD pattern of the pristine SWCNTs (Figure 3c) are shifted to $23.97°$ and $27.92°$, respectively, in the pattern of the ZnO–SWCNT composite (Figure 3a,d). This result is attributed to the change in crystallinity of carbon because of chemical reaction with the ZnO nanorods during heating. That is, the ZnO nanoparticles can interact with SWCNTs, producing covalent bonding such as ZnO–SWCNT or Zn–OOC–SWCNT or attaching through van der Waals forces [28].

The elemental composition and chemical states of the constituent elements in the prepared nanocomposite were evaluated by XPS. Figure 4a presents a high-resolution survey spectrum of the composite, indicating the presence of Zn, O, and C. Figure 4b shows the high-resolution XPS spectrum

corresponding to the Zn 2p core level of the ZnO–SWCNT composite. The wide and asymmetric outline is separated into two distinct curves, which indicates the existence of two types of $Zn^{2+}$ ions in different chemical environments. The $2p_{3/2}$ and $2p_{1/2}$ peaks at 1021.44 and 1044.82 eV correspond to Zn in the $Zn^{2+}$ state in the ZnO–SWCNT composite. The lack of a distinct bud peak (hump) associated with the main $Zn^{2+}$ peak implies that the zinc ions maintain their divalent state in the whole sample. No other oxidation state of Zn was indicated.

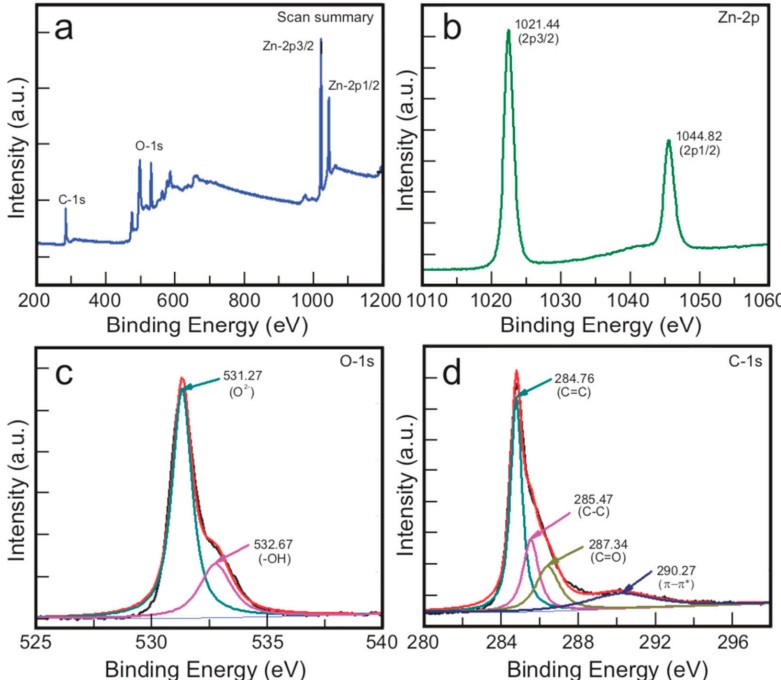

**Figure 4.** XPS spectra: (**a**) survey spectrum of the prepared ZnO–SWCNT nanocomposite and high-resolution spectra showing (**b**) the Zn-$2p_{3/2}$ and Zn-$2p_{1/2}$ peaks, (**c**) the O-1s core level, and (**d**) the C-1s core level of the ZnO–SWCNT nanocomposite.

The O-1s core-level spectrum is presented in Figure 4c. The large asymmetric features demonstrated by these profiles are divided into two distinct parts, indicating the occurrence of two nonequivalent chemical environments around oxygen ions. The comparatively robust component (O-1s) at lower binding energy (531.27 eV) represents $O^{2-}$ ions bound with $Zn^{2+}$ ions of the wurtzite form at tetrahedral positions. The other peak at higher binding energy (532.67 eV) represents surface-chemisorbed –OH groups, indicating the existence of weakly bonded oxygen atoms near the surface region of the ZnO nanorods [1,4]. The C-1s core-level spectrum (Figure 4d) reveals peaks at 284.76, 285.47, 287.34, and 290.27 eV, which are attributed to C=C, C–C, and C=O bonds and the $\pi$–$\pi^*$ transition of C=C, respectively, in the ZnO–SWCNT nanocomposite [28,41].

The nanostructure of the ZnO–SWCNT nanocomposite was characterized by HR-TEM; the corresponding micrographs are displayed in Figure 5. Relatively lower-magnification TEM images are presented in Figure 5a,b. An intimate association of ZnO with SWCNTs is evident. Most of the ZnO particles are rod-shaped, with smooth surfaces. Their width and length range from 40 to 60 nm and from 50 to 300 nm, respectively. Higher-magnification images in which SWCNTs are discernible are shown in Figure 5c,d. In the magnified images, the very thin SWCNTs are observed as impressions or markings on the ZnO surfaces. The ZnO and SWCNTs are in intimate contact, leading to the formation of a ZnO–C heterojunction, which is essential for the efficient photocatalytic performance of the semiconductor nanocomposite [32].

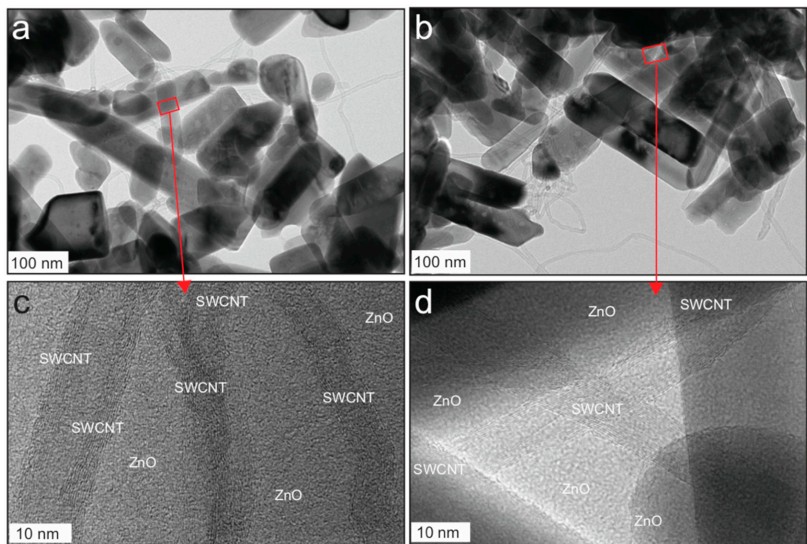

**Figure 5.** (**a**,**b**) Low-magnification HR-TEM images of the ZnO–SWCNT nanocomposite, and (**c**,**d**) high-resolution images corresponding to the region marked in images (**a**,**b**), respectively.

## 2.3. Optical Properties of the Nanocomposite, Pristine ZnO, and Pristine SWCNTs

UV–vis spectra of the ZnO–SWCNT nanocomposite were recorded for the composite dispersed in ethanol in very dilute concentrations (0.008 mg/mL). Samples of pristine ZnO and SWCNTs were prepared in ethanol with the same concentration and scanned. The absorption spectra corresponding to these mixtures are presented in Figure 6. Compared with the pristine ZnO or pristine SWCNTs, the ZnO–SWCNT composite displays robust absorption over the examined wavelength range. The ZnO peak at 384 nm in the spectrum of the composite is red-shifted compared with the peak in the spectrum of pristine ZnO (376 nm). This shift is attributed to enhanced light absorption due to the chemical bonding of ZnO with C. Furthermore, multiple scattering of light by vaguely oriented ZnO nanorods in the ZnO–SWCNT nanocomposite can also enhance light absorption [28]. The UV–vis absorption spectra indicate that the synthesized photocatalyst can absorb visible light to produce a substantial number of electron–hole pairs. The apparent bandgap energy was also calculated using Tauc plot from the UV-vis absorbance data. The Tauc plot (Figure S3) shows a significant decrease in the theoretical bandgap of ZnO in the composite.

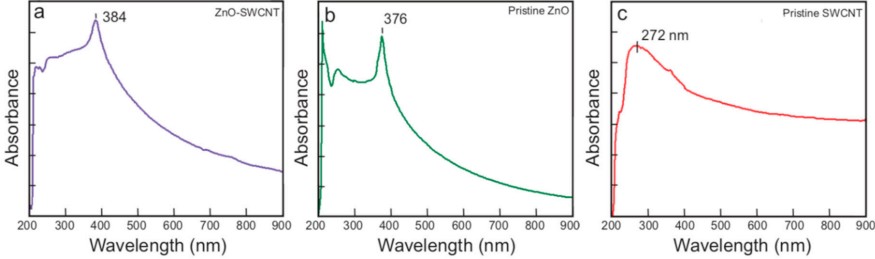

**Figure 6.** UV–vis absorption spectra of (**a**) a ZnO–SWCNT nanocomposite, (**b**) pristine ZnO nanorods, and (**c**) pristine SWCNTs.

## 2.4. Thermal Properties of the Nanocomposite

A thermal analyzer was used to examine the thermal stability of the ZnO–SWCNT nanocomposite under a $N_2$ atmosphere between 26 and 800 °C, maintaining a 10 °C min$^{-1}$ heating rate. Figure 7 reveals the thermogravimetric analysis (TGA) and differential scanning calorimetry (DSC) curves of the nanocomposite, pristine SWCNTs, and pristine ZnO. The pristine SWCNTs were heated under atmospheric air during the analysis. Pristine ZnO exhibited a slight mass loss beginning at

approximately 100 °C as a result of evaporation of water adsorbates. The mass loss at temperatures above 200 °C is attributed to the dissociation of residual acetates. No further thermal degradation was observed as the temperature was increased to 800 °C under the $N_2$ environment.

Pristine SWCNTs degraded slowly with increasing temperature in the temperature range between 100 °C and 400 °C in air and very rapidly thereafter. Nearly 96% of the original content was decomposed during heating to 570 °C. By contrast, no substantial degradation of the ZnO–SWCNT composite was observed during heating to 500 °C. As the temperature was gradually increased to 800 °C, degradation of the composite appeared to be negligible compared with that of the pristine SWCNTs, but similar to that of the pristine ZnO. Thus, the thermal analysis reveals that the prepared nanocomposite is very stable compared with the pristine SWCNTs. The increased resistance to thermal decomposition indicates strong chemical interaction of ZnO with the SWCNTs [28].

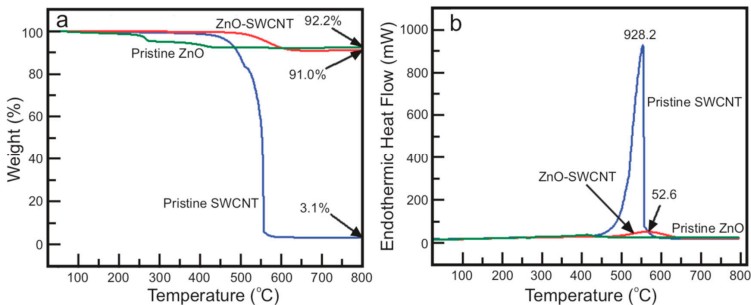

**Figure 7.** Thermal analysis of the ZnO–SWCNT nanocomposite, the pristine ZnO, and the pristine SWCNTs: (**a**) TGA curve and (**b**) DSC curve.

## 2.5. Photocatalytic Performance

The catalytic performances of the pristine ZnO nanorods, the pristine SWCNTs, and the ZnO–SWCNT nanocomposite, prepared through the same route and under the same conditions, were studied in an outdoor environment through the photocatalytic decomposition of MB under visible-light irradiation. Transmission of UV light was restricted by a UV cut-off filter such that only visible and near-infrared light was allowed to enter the reaction vial. The photocatalytic performances of all of the samples are shown in Figure 8. The decomposition rate of MB by the pristine ZnO nanorods or pristine SWCNTs appears prolonged, the reaction being chiefly assigned to the photosensitization of the dye. By contrast, the ZnO–SWCNT nanocomposites show remarkably higher photocatalytic performance (Figure 8a) than the pristine ZnO (Figure 8b) and the pristine SWCNTs (Figure 8c). The degradation rate constants for the use of ZnO-SWCNT, pristine ZnO, and pristine SWCNTs as photocatalyst were also calculated, and the values are 0.0323, 0.0068 and 0.0005 min$^{-1}$, respectively. These findings imply that compositing SWCNTs with ZnO nanorods promotes the separation of photoinduced electron–hole pairs, which is critical for enhancing the photodegradation ability of the substrate [39].

We examined the effect of photocatalyst dose on the efficiency of decomposition of MB by varying the amount of photocatalyst from 0.05 to 0.130 g in 100 mL of the MB solution (0.25 mg mL$^{-1}$, $7.9 \times 10^{-4}$ M in water). The efficiency of the photocatalyst increased with increasing photocatalyst dose, attaining the highest value at the dose of 0.130 g. The degradation efficiency began diminishing as the photocatalyst dose was increased more than 0.130 g. An optimum dose of the photocatalyst is required for efficient photocatalytic performance because numerous active sites on the surface of a photocatalyst or its surface area increases with increasing photocatalyst dose. This interpretation implies that the amounts of superoxide and hydroxyl radicals, which play a critical role in the photodegradation reaction, also increase. However, the degradation efficiency decreases above the optimum dose of photocatalyst because of the increasing opacity of the suspension. The opacity prevents the transmission of light and makes the photocatalyst surface unavailable for light absorption and the generation of charge carriers.

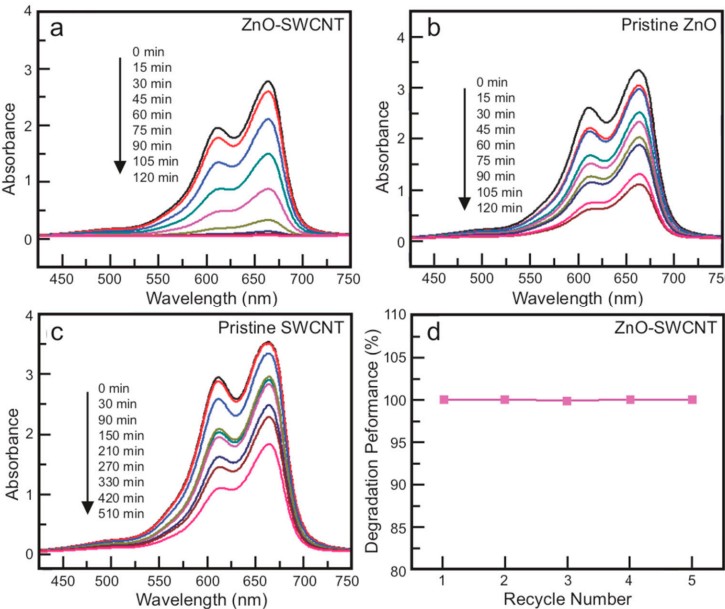

**Figure 8.** UV–vis absorption spectra of a methylene blue (MB) solution as a function of time in the presence of (**a**) the ZnO–SWCNT nanocomposite, (**b**) pristine ZnO nanorods, and (**c**) pristine SWCNTs. (**d**) Recycling performance of the ZnO–SWCNT nanocomposite.

The initial concentration of MB solution was also found to affect its photodegradation. We discerned that the photodegradation of the dye solution was less extensive when the dye concentration was higher. This observation is attributed to the difficulty associated with forming reactive oxygen species (ROS)—specifically, superoxide radicals ($^\bullet O_2^-$) and hydroxyl radicals ($^\bullet OH$), which we speculate are the main species responsible for the photodegradation of the dye. Greater coverage of the active sites of the photocatalyst by the MB solution makes the formation of ROS more difficult and, hence, reduces the catalyst performance. Furthermore, in the case of a high concentration of dye, more of the incident light is absorbed by dye molecules than by the reactive sites of the photocatalyst, which hampers the generation of ROS and, ultimately, the photocatalytic performance [1,12,13]. The recycling efficiency of the photocatalyst is displayed by Figure 8, which demonstrates no significant loss in performance of the catalyst until the fifth cycle. There is only one percent loss of activity, i.e., 99% of the activity of the photocatalyst remains intact even after five times recycling.

The photocatalytic efficiency of ZnO-SWCNT nanocomposite was compared with that of previously reported composites consisting of ZnO and CNTs. The comparative results are presented in Table 1. The photocatalytic efficiency of ZnO-SWCNT composite in terms of the rate constant was also compared with those of previously reported composites. The comparative results are presented in Table 2. The results show how our composite is a more effective photocatalyst. The results depict that our ZnO-SWCNT is not only effective, but also a unique photocatalyst involving the use of SWCNTs.

**Table 1.** Comparison of photocatalytic efficiency of composites involving ZnO-CNTs.

| Composite | Pollutant | Pollutant Concentration | Composite Doze | Degradation (%) | Band Energy (nm) | Degradation Time (min) | Ref. |
|---|---|---|---|---|---|---|---|
| ZnO-MWCNT | MB | 0.01 mmol (100 mL) | 0.1 g | 100 | 365 | 150 min under 250 W lamp | [16] |
| ZnO-MWCNT | Acetaldehyde | 0.9 mmol | 0.5 g/L | 71 | 365 | 120 min under UV | [26] |
| ZnO-MWCNT | MB | 4 mg/L | 100 mg/ 250 mL | 93 | N/A | 40 min under UV | [27] |
| ZnO-SWCNT | MB | 0.25 mg/mL (100 mL) | 130 mg | 100 | 384 | 120 min (Sunlight) | Our work |

**Table 2.** Comparison of photocatalytic performance in terms of rate constants.

| Composite | Rate Constant (k) (min$^{-1}$) | Reference |
|:---:|:---:|:---:|
| Fe-Cd-doped ZnO | 0.01153 | [1] |
| ZnO-Ag | 0.0295 | [6] |
| Cd-doped ZnO-MWCNT | 0.007 | [14] |
| ZnO-rGO | 0.025 | [32] |
| Ni-doped ZnO | 0.017 | [35] |
| ZnO-SWCNT | 0.0323 | Our work |

### 2.6. Proposed Mechanism of Photodegradation

A proposed mechanism of the photodecomposition of MB by the ZnO–SWCNT photocatalyst is presented in Figure 9. When the MB solution with the ZnO–SWCNT photocatalyst is irradiated with sunlight with energy ($hv$) greater than or equal to the activation energy ($E_a$), electrons from the occupied valence band (VB) of ZnO jump to the vacant conduction band (CB). Hence, electrons ($e^-$) occupy the conduction band and holes ($h^+$) occupy the valence band of ZnO. These electron–hole pairs then migrate to the ZnO surface [42]. Because the SWCNTs are good electronic conductors, the photogenerated electrons are transferred to the SWCNTs quickly from the interface, whereas the holes remain in ZnO [15,27]. These electrons combine with oxygen (water contains some dissolved oxygen) to generate superoxide radicals ($^{\bullet}O_2{}^-$) instead of returning to the VB or recombining with holes. The holes attack waters or OH$^-$ ions and produce $^{\bullet}$OH radicals [28]. Thus, the SWCNTs separate the photogenerated electron–hole pairs and prevent them from recombining; the electron–hole pairs thus remain available to participate in redox reactions.

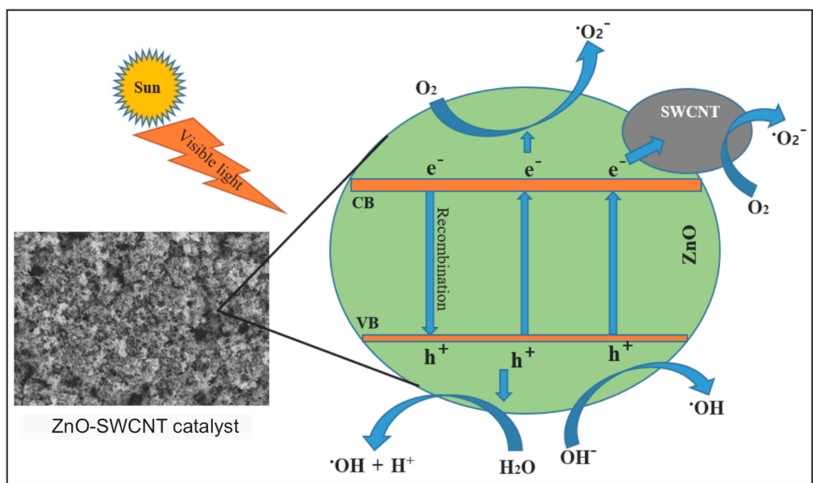

**Figure 9.** Schematic of the generation of active species by the ZnO–SWCNT photocatalyst for MB degradation.

The superoxide radicals ($^{\bullet}O_2{}^-$) react further to produce powerful oxidizing agents, i.e., hydroxyl radicals. The hydroxyl radicals instantly attack dye molecules in contact with the ZnO–SWCNT surface and generate transitional compounds or intermediates. Finally, the intermediates are transformed into innocuous products such as water, CO$_2$, and mineral acids [42]. The mechanism of decomposition of the organic pollutant MB by the ZnO–SWCNT photocatalyst in the presence of sunlight is summarized as follows:

$$\text{SWCNT-ZnO} + hv \rightarrow \text{SWCNT-ZnO } (e^-{}_{(CB)}) + \text{SWCNT-ZnO } (h^+{}_{(VB)}) \tag{1}$$

$$\text{SWCNT-ZnO } (h^+{}_{(VB)}) + \text{H}_2\text{O} \rightarrow \text{SWCNT-ZnO} + \text{H}^+ + {}^{\bullet}\text{OH} \tag{2}$$

$$\text{SWCNT-ZnO } (h^+{}_{(VB)}) + \text{OH}^- \rightarrow \text{SWCNT-ZnO} + {}^{\bullet}\text{OH} \tag{3}$$

$$\text{SWCNT-ZnO } (e^-_{(CB)}) + O_2 \rightarrow \text{SWCNT-ZnO} + {}^\bullet O_2{}^- \tag{4}$$

$$O_2 + H^+ \rightarrow HO_2{}^\bullet \tag{5}$$

$$HO_2{}^\bullet + HO_2{}^\bullet \rightarrow H_2O_2 + O_2 \tag{6}$$

$$\text{SWCNT-ZnO } (e^-_{(CB)}) + H_2O_2 \rightarrow {}^\bullet OH + OH^- \tag{7}$$

$$H_2O_2 + {}^\bullet O_2{}^- \rightarrow {}^\bullet OH + OH^- + O_2 \tag{8}$$

$$H_2O_2 + h\nu \rightarrow 2\,OH^\bullet \tag{9}$$

$$\text{SWCNT-ZnO } (h^+_{(VB)}) + \text{Organic pollutant} \rightarrow R^\bullet \text{ (Intermediates)} \tag{10}$$

$$\text{Organic pollutants} + OH^\bullet \rightarrow R^\bullet \tag{11}$$

$$R^\bullet \rightarrow CO_2 + H_2O \tag{12}$$

## 3. Experimental Section

### 3.1. Chemicals

SWCNTs (>90%, outer diameter: 1–2 nm, US Research Nanomaterials, Inc., 3302 Twig Leaf Ln, Houston, TX, USA), zinc acetate dihydrate (Sigma-Aldrich, 99% purity, St. Louis, MO, USA), MB (Alfa Aesar, high purity, biological stain, Heysham, Lancashire, UK), and ethanol (Sigma-Aldrich, 99.5% purity) were used as received. Distilled water was used to prepare MB solutions.

### 3.2. Characterization

Morphological characteristics of the prepared nanocomposite were investigated by field-emission scanning electron microscopy (FE-SEM, Carl Zeiss, SUPRA 40 VP, Oberkochen, Germany) and high-resolution transmission electron microscopy (HR-TEM, JEM 2010, JEOL, Japan). The structural characteristics were investigated by X-ray diffraction (Multi-Purpose High Performance X-ray Diffractometer, X'pert Pro Powder, PANalytical, The Netherlands). Spectroscopic features were studied by UV–vis spectrophotometry (PerkinElmer Lambda 25, Ayer Rajah, Singapore) and X-ray photoelectron spectroscopy (XPS, VG Multilab 2000, Waltham, MA, USA). Thermal analysis was performed with thermal analysis equipment (SDT Q600 V20.9 Build 20, New Castle, DE, USA).

### 3.3. Fabrication of the ZnO–SWCNT Nanocomposite

A sonicator bath was used to prepare a solution of zinc acetate dihydrate in ethanol. Recrystallization of zinc acetate onto SWCNTs was carried out in a glass beaker, and the obtained product was calcined in a compact muffle-type furnace (KSL-1100X-S-UL-LD, Richmond, CA, USA). In a typical experiment, zinc acetate dihydrate (6.0 g) was transferred into ethanol (100 mL) in a glass beaker and the resultant mixture was placed in a sonicator bath for 1 h at ambient temperature (~20 °C) to ensure homogeneous mixing of the constituents. Then, 100 mg of SWCNTs was added to the solution, and the resultant mixture was subjected to magnetic stirring for 1 h. The magnetic stirrer was removed from the beaker, and the mixture was allowed to stand undisturbed for 5 h. Crystals started to form immediately after the stirrer was removed; however, complete and uniform recrystallization around the SWCNTs required approximately 5 h. The zinc acetate crystal–SWCNT composite was separated from the ethanol via vacuum filtration. The composite was then desiccated at 60 °C in a preheated oven for 1 h. The dried zinc acetate crystals–SWCNT composite was placed in a quartz crucible with a lid. The crucible was then inserted into a sealed stainless-steel chamber with an oxygen-free, high-conductivity copper gasket seal (SUS 314) and subjected to furnace heating at 500 °C for 30 min to fabricate the ZnO–SWCNT nanocomposite. The average molar proportion of constituents in the ZnO–SWCNT composite was 1.96:1 (ZnO:SWCNTs), as computed from the masses of the SWCNTs and the ZnO–SWCNT nanocomposite. The average ratio was calculated by repeating the experiment

five times. We found that nanocomposites with high photocatalytic efficiency were formed only when they were synthesized by dispersing the same amount of zinc acetate dihydrate (6.0 g) in 100 mL of ethanol and 100 mg of SWCNTs, or when multiples of these amounts were used. In other cases, either excess deposition of ZnO nanorods upon the SWCNTs or some bare CNTs were observed.

### 3.4. Preparation of Pristine ZnO Nanorods

Pristine ZnO nanorods were prepared using zinc acetate dihydrate and ethanol via the method described in Section 2.3 to compare the photocatalytic abilities of the ZnO–SWCNT nanocomposite and pristine ZnO.

### 3.5. Photocatalysis Experiments

The ZnO–SWCNT nanocomposite (130 mg) was transferred into an MB solution (100 mL, $7.9 \times 10^{-4}$ M in distilled water). The mixture was subjected to bath sonication for 1 h and then allowed to stand in the dark for 1 h to attain equilibrium. The decomposition of MB via photocatalysis was performed under uninterrupted sunlight for 120 min within the temperature range 8–11 °C (10:30 am to 12:30 pm in the outdoor environment) in February (the average radiation intensity is 5.37 kWh/m$^2$/day in February). The transmission of UV light into the reaction vial was prevented by a UV cut-off filter such that only visible and near-infrared wavelengths could affect the reaction. The degradation of the MB solution was accessed via absorbance measurements (UV–vis spectrophotometer). The recycling performance of the ZnO–SWCNT nanocomposite was assessed by collecting the ZnO–SWCNT nanocomposite, drying it at 100 °C for 10 min, and reusing it; this process was repeated for five cycles. Control experiments were performed with pristine ZnO nanorods prepared via the same method and with pristine SWCNTs.

## 4. Conclusions

A natural-sunlight-driven, highly efficient ZnO–SWCNT photocatalyst was synthesized via a one-pot recrystallization method followed by thermal decomposition. We assessed its photocatalytic performance by testing its ability to photocatalyze the decomposition of MB as a function of time and testing its recyclability. Its morphological characteristics demonstrate that the prepared nanocomposite consists of SWCNTs covered with ZnO nanorods in random orientation. The absorption of visible light by ZnO is substantially enhanced, and the bandgap of ZnO is red-shifted because of the formation of a chemical bond with the SWCNTs. The XPS results indicate the presence of divalent $Zn^{2+}$ ions in chemical combination with $O^{2-}$ and carbon. The catalytic efficiency of the ZnO–SWCNT photocatalyst is greater than the efficiencies of the pristine ZnO and the pristine SWCNTs, which is attributed to the chemical bonding between the ZnO and the SWCNTs. The chemical bonding promotes light absorption and inhibits the recombination of electron–hole pairs generated during catalysis. The proposed alternative mechanism reflects the improvement of the catalytic performance of the nanocomposite. The synthesized photocatalyst can efficiently decompose organic pollutants such as MB into green products in the natural environment.

**Supplementary Materials:** The following are available online at http://www.mdpi.com/2073-4344/9/6/498/s1. Figure S1: UV-vis absorption spectra of MB solution as a function of time in the presence of the ZnO–SWCNT nanocomposite in the range of 200-900 nm wavelength, Figure S2: UV-vis absorption spectra of MB solution as a function of time without the use of catalyst, Figure S3: A Tauc curve for the determination of band-gap of the photocatalyst.

**Author Contributions:** K.P.S. designed and performed the experiments; J.R.H. supervised the research work; I.L. made arrangements of reagents, materials and analysis tools; K.P.S. and I.L. analyzed data and K.P.S. wrote the paper under the guidance of J.R.H.; M.A.I. and M.A.H. analyzed data and read the manuscript thoroughly to improve it.

**Acknowledgments:** This work was supported by the Korean Government, NRF–2018R1A2B6006155.

**Conflicts of Interest:** The authors declare no conflict of interest.

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
