# Peer review of "Solar-Light-Driven Efficient ZnO–Single-Walled Carbon Nanotube Photocatalyst for the Degradation of a Persistent Water Pollutant Organic Dye"

_catalysts, doi:10.3390/catal9060498_

Reviewer 1 Report

The authors report on the preparation of ZnO-SWCNT hybrid catalyst toward methylene blue degradation using ambient sunlight. Blank experiments have been performed, experimental conditions are adequately prepared, and indeed the activity between ZnO and SWCNT seems synergistic. In my view the manuscript is very well prepared and does not require any changes, therefore I suggest publication in Catalysts in its present form.

Author Response

Response to Reviewer-1’s comments

The authors report on the preparation of ZnO-SWCNT hybrid catalyst toward methylene blue degradation using ambient sunlight. Blank experiments have been performed, experimental conditions are adequately prepared, and indeed the activity between ZnO and SWCNT seems synergistic. In my view the manuscript is very well prepared and does not require any changes, therefore I suggest publication in Catalysts in its present form.

 Author Reply: We are thankful to you for this encouraging comment.

Reviewer 2 Report

1. Development of methods of obtaining ZnO nanostructures with controlled properties, such as shape or size (specific surface area), enables their repeatable application, as especially photocatalysts. At present, the synthesis of nanomaterials is at a new stage of development, where it is a common requirement to control the properties of the obtained nanostructures, their repeatability and reproducibility. I encourage the authors to underline this fact in the “Introduction” and add a sentence of a comment about the development of such methods of obtaining nanostructures, from nanoparticles to nanorods, with the following publications serving as examples:
- Size control mechanism of ZnO nanoparticles obtained in microwave solvothermal synthesis, 2018, Nanotechnology, 29, 065601.
- Morphology and size controlled synthesis of zinc oxide nanostructures and their optical properties, J Mater Sci: Mater Electron, 2018, 29, 9339.
- Size Control of ZnO Nanorods Using the Hydrothermal Method in Conjunction with Substrate Rotation, Journal of Nanoscience and Nanotechnology, Volume 17, Number 11, November 2017, pp. 7952-7956(5).

2. The description of the “Fabrication of the ZnO–SWCNT nanocomposite” part is not understandable. The authors first describe obtaining the ZnO–SWCNT composite at the temperature of 1100°C without providing the parameters (calcination duration, gaseous environment) (115-116). The subsequent text fragment describes obtaining the ZnO–SWCNT composite at the temperature of 500°C (126-127). At which temperature were the target ZnO–SWCNT composite samples obtained? Why were two calcination temperatures provided?

3. How did the authors calculate the average molar proportion of constituents in the ZnO–SWCNT composite being 1.96:1? In the synthesis, 6 g (0.0273 mol) of zinc acetate dehydrate and 100 mg (0.00833 mol) of SWCNTs (132-133) were used, so the average molar proportion of constituents in the ZnO–SWCNT composite should theoretically be 3.28:1. What is the cause of the difference?

4. Based on the SEM photographs (Figure 1) please supplement the sentence: “The ZnO nanorods have various lengths and diameters, with random  spatial orientation. (159-160)“ with indicative ranges of lengths and diameters of the obtained ZnO nanorods.

5. The authors must become familiar with ISO_22309_2011(E) standard, which describes the manner of correct performance of EDS measurement. I think that the EDX results in this form contribute nothing to this paper and confirm what is already known, i.e. that the sample contains C, O, Zn (166-171). I suggest that the authors think about the sense of including these results in their paper.

6. The sentence: “The nanoscale size range of ZnO particles in the synthesized composite is confirmed by the precise expansion of the linewidth of the ZnO XRD peaks (180-181)” does not tell the truth. ZnO NPs peaks are narrow in Figure 3a and 3b, which rather indicates obtaining ZnO microparticles. Please determine the size of ZnO crystallites based on XRD data (Figure 3) using e.g. the Scherrer’s formula.

7. Why were two different gases (nitrogen and air) used for thermal stability tests (259-260; 262-263)? What was the impact on the thermal stability results obtained?

8. One significant parameter defining the photocatalyst is missing, namely specific surface area. Are the authors capable of performing an analysis of the specific surface area of the samples?

Author Response

Response to Reviewer-2’s comments

Development of methods of obtaining ZnO nanostructures with controlled properties, such as shape or size (specific surface area) enables their repeatable application, as especially photocatalysts. At present, the synthesis of nanomaterials is at a new stage of development, where it is a common requirement to control the properties of the obtained nanostructures, their repeatability and reproducibility. I encourage the authors to underline this fact in the “Introduction” and add a sentence of a comment about the development of such methods of obtaining nanostructures, from nanoparticles to nanorods, with the following publications serving as examples:

- Size control mechanism of ZnO nanoparticles obtained in microwave solvothermal synthesis. 2018, Nanotechnology. 29, 065601.

- Morphology and size controlled synthesis of zinc oxide nanostructures and their optical properties, J Mater Sci: Mater Electron, 2018, 29. 9339.

- Size Control of ZnO Nanorods Using the Hydrothermal Method in Conjunction with Substrate Rotation, Journal of Nanoscience and Nanotechnology, 17 (11), 2017, pp.7952-7956(5).

Author Reply: We are thankful to you for the above suggestion. We revised to include the research in the text (lines 85-87) and Refs. 38-40 to emphasize the control of properties of the nanostructures for their efficient and repeatable photocatalytic performance.

2. The description of the “Fabrication of the ZnO-SWCNT nanocomposite” part is not

understandable. The authors first describe obtaining the ZnO-SWCNT composite at the temperature of 1100oC without providing the parameters (calcination duration, gaseous environment) (115-116). The subsequent text fragment describes obtaining the ZnO-SWCNT composite at the temperature of 500oC (126-127). At which temperature were the target ZnO-SWCNT composite samples obtained? Why were two calcination temperatures provided?

Author Reply: We thank you for pointing out our mistake. In fact, we meant to mention the furnace name, i.e., 1100°C Compact Muffle Furnace (line 118). We revised to remove 1100°C (indicating the maximum temperature capacity of the furnace) to avoid confusion. The ZnO-SWCNT samples were obtained at 500°C as expressly mentioned in line 129.

3. How did the authors calculate the average molar proportion of constituents in the ZnO-SWCNT composite being 1.96:1? In the synthesis, 6 g (0.0273 ma) of zinc acetate dehydrate and 100 mg (0.00833 mol) of SWCNTs (132-133) were used, so the average molar proportion of constituents in the ZnO-SWCNT composite should theoretically be 3.28:1. What is the cause of the difference?

Author Reply: We calculated the average molar ratio of the composite assuming that whole amount of the SWCNTs (i.e., 100 mg or 0.00833 mol) remains intact in the composite as it is not soluble in ethanol. Since some amount of zinc acetate remains in mother liquor even after recrystallization, its amount in the composite is less. Keeping in mind the above assumption, we calculated the molar ratio as follows:

Mass of SWCNTs = 100 mg = 0.1 g = 0.00833 mol

Recorded yield of composite = 1.4293 g

Mass of ZnO in the composite = (1.4293 – 0.1) g = 1.3293 g

Number of moles of ZnO = 1.3293/81.39 = 0.01633 mol

Molar ratio = 0.01633:0.00833 = 1.96:1

4. Based on the SEM photographs (Figure 1) please supplement the sentence: "The ZnO nanorods have various lengths and diameters, with random spatial orientation. (159-160)" with indicative ranges of lengths and diameters of the obtained ZnO nanorods.

Author Reply: We thank you for pointing out this issue. The width and length of the ZnO rods as evaluated through FE SEM images were in the ranges of 40 to 60 nm and from 50 to 300 nm, respectively. More precise measurements were taken through HR TEM. Thus, we expressly mentioned the width and length range of the ZnO rods in HR TEM description (lines 236-237).

5. The authors must become familiar with ISO_22309_2011 (E) standard, which describes the

manner of correct performance of EDS measurement. I think that the EDX results in this form

contribute nothing to this paper and confirm what is already known, i.e. that the sample contains

C, O, Zn (166-171). I suggest that the authors think about the sense of including these results in

their paper.

Author Reply: We thank you for the very natural suggestion. As you mentioned, the present EDX results contribute little to this paper. However, in our past experiences, many readers would like to see the EDX to confirm the composition and credibility of the experimentation. In Figure 2, we tried to illustrate the length, diameter and arrangement of the nanorods (around SWCNTs in space) through the schematic diagram (Fig. 2c) with the EDX. The EDX is presented to support the schematic diagram.

6. The sentence: "The nanoscale size range of ZnO particles in the synthesized composite is

confirmed by the precise expansion of the linewidth of the ZnO XRD peaks (180-181)" does not

tell the truth. ZnO NPs peaks are narrow in Figure 3a and 3b, which rather indicates obtaining

ZnO microparticles. Please determine the size of ZnO crystallites based on XRD data (Figure 3)

using e.g. the Scherrer’s formula.

Author Reply: We thank you for the correction and suggestion. According to your correction, we removed the sentence from this manuscript. Also, we revised to include the calculated size of the ZnO crystallites based on XRD data according to your suggestions (lines 184-186).

7. Why were two different gases (nitrogen and air) used for thermal stability tests (259-260; 262-263)? What was the impact on the thermal stability results obtained?

Author Reply: Our objective was to evaluate the thermal stability of the prepared composite. We assumed that if the prepared composite undergoes partial combustion in the air forming solid product by chemical decomposition, it could affect the photocatalytic performance. Nitrogen gas was used to provide an inert atmosphere. Since SWCNTs are elemental carbon, there is no thermal degradation (into simpler products) in an inert atmosphere. In the atmosphere of air, its amount degrades through combustion producing gaseous carbon compounds (thermal oxidation). Furthermore, heating at a higher temperature may result in the formation of nitride instead of decomposition. Therefore, the air was used in the case of SWCNTs.

8. One significant parameter defining the photocatalyst is missing, namely specific surface area.

Are the authors capable of performing an analysis of the specific surface area of the samples?

Author Reply: We thank you for the very natural question. As you mentioned the specific surface area is one of the parameters defining the photocatalyst. In the present paper, however, we focused on the formation of ZnO-C heterojunction (as illustrated in Figure 5) and a marked red shift in UV-vis spectra of the composite (Figure 6), which provide the necessary support for the enhanced photocatalytic performance. The formation of chemical bonds is also confirmed by XPS results. We did not measure the specific surface area of the composite. In fact, we are not sure how much specific surface area is modified by the formation of the composite in the current work. We have a plan to do further research for the ZnO-SWCNT composite with different size and shape of ZnO (nanorods, nanosheets, and nanospheres), which requires measurement of the specific surface area. Thus, we left it for a future research.

Reviewer 3 Report

The manuscript by Sapkota and co-authors is not suitable for publication in Catalysts. 

The manuscript lacks of novelty. Literature is plenty of papers dealing with ZnO couple with (SMW)CNT and photocatalytic degradation of MB. 

Furthermore using organic dyes and visible light is not recommended to show photocatalytic activity. See DOI: 10.1016/j.cattod.2013.12.019

Author Response

Response to Reviewer-3’s comments

The manuscript by Sapkota and co-authors is not suitable for publication in Catalysts. The manuscript lacks of novelty. Literature is plenty of papers dealing with ZnO couple with (SMW)CNT and photocatalytic degradation of MB. Furthermore using organic dyes and visible light is not recommended to show photocatalytic activity See DOI: 10.1016/j.cattod.2013.12.019.

Author Reply: We do not agree with your comments. The novelty of our work is the fabrication of an efficient photocatalyst (ZnO–SWCNT nanocomposite) through a straightforward one-pot–two-chemical recrystallization followed by thermal decomposition. The synthesized photocatalyst is an effective and recyclable agent for the decomposition of an MB solution. Our method is not found to be reported in the literature. Furthermore, the photocatalyst synthesized by this method is more efficient in comparison to those synthesized through other methods reported in the literature (see Table 1 and 2). We would be glad if you provide more concrete and specific grounds for your comments. The comments are too conclusive and short to understand what grounds you have and to make our response.

Reviewer 4 Report

An article entitled Solar-light-driven efficient ZnO–SWCNT photocatalyst for the degradation

of a persistent water pollutant organic dye describes the photocatalytic activity of

ZnO-based nanocomposite under the sunlight.

The introduction section is well-written and provides a piece of sufficient information

from the field, however, in the other parts of paper, some key factors are missing.

Please find some comments below:

1.            The Authors report that the photodegradation of the dye occurs, without showing any credible evidence, such as the measurement of total organic carbon (TOC) in treated solution. The only process that Authors observe is the decolorization of the dye, which does not mean mineralization to CO2 and H2O. For this reason all section 3.6. Proposed mechanism of photodegradation it is not significant.

2.            The Authors do not give such significant parameters of the process as the intensity of radiation and temperature of the solution. Thermal decomposition of the dye cannot be ruled out.

3.            What is more, the Authors of the work should determine the constant rate of decolorization, because it is a parameter that can be compared, and comment on the received values in the light of other photocatalysts reported in the literature.

Author Response

Response to Reviewer-4’s comments

An article entitled Solar-light-driven efficient ZnO-SWCNT photocatalyst for the degradation of a persistent water pollutant organic dye describes the photocatalytic activity of ZnO-based nanocomposite under the sunlight. The introduction section is well-written and provides a piece of sufficient information from the field, however, in other parts of paper, some key factors are missing.

1. The Authors report that the photodegradation of the dye occurs, without showing any credible evidence, such as the measurement of total organic carbon (TOC) in treated solution. The only process that Authors observe is the decolorization of the dye, which does not mean mineralization to CO2 and H2O. For this reason, all section 3.6. Proposed mechanism of photodegradation is not significant.

Author Reply: We thank you for the useful remark, which we have not considered. Due to our limited research facility, we were able to measure the removal rates of TOC for a few samples.

When MB was decomposed in the presence of ZnO–SWCNT for 105 min at the same conditions, 96.9% of TOC was decomposed. This percentage is similar to that of UV/vis absorption measurement. In agreement with you, measuring a removal rate of COD or TOC is a reliable way to test water quality. The UV/vis absorbance technique is also an analytical technique to measure a concentration of the sample in a chemical solution. Previously, many reports [R1~R5] have presented the UV/vis absorbance results to monitor the chemical reaction. Measurements of COD, TOC, and COD/TOC ratio would be very useful to understand the catalytic mechanism and efficiency in more detail. Thus, we have seriously tried to follow the suggestion, but at the end, we concluded that it would go way beyond the scope of this paper, that is, it would require much more additional and systematic work with various samples. Hence, we could not add enough data in the present paper now and left it for a future research.

[R1] J. Am. Chem. Soc. 2016, 138, 5978−5983.

[R2] Angew. Chem. Int. Ed. 2016, 55, 5342–5345.

[R3] ACS Catal. 2016, 6, 1744−1753.

[R4] ACS Catal. 2016, 6, 3594−3599.

[R5] Nanoscale, 2016, 8, 365–377

Our proposed mechanism is mainly based on the formation of heterojunction (via chemical bond) between ZnO and C that suppresses the recombination of photogenerated electron-hole pairs, whereby electrons and holes become available to participate in redox reactions. These ideas are also supported by many publications.

2. The Authors do not give such significant parameters of the process as the intensity of radiation and temperature of the solution. Thermal decomposition of the dye cannot be ruled out.

Author Reply: As stated in the section 2.5 Photocatalysis experiments, we have performed the experiment in normal environmental conditions within the outside temperature range of 8-11°C. No extra heat was supplied to the dye solution before exposing it to the sunlight in the presence of photocatalyst. Furthermore, we also evaluated the self-decomposition of dye (of the same concentration and volume) in the absence of photocatalyst in sunlight for 120 minutes. In this case, we did not find a noticeable change in concentration of the dye solution (it was evaluated through UV-vis absorbance measurement). After having such observations, we assumed, thermal decomposition of the dye did not play a significant role.

3. What is more, the Authors of the work should determine the constant rate of decolorization, because it is a parameter that can be compared, and comment on the received values in the light of other photocatalysts reported in the literature.

Author Reply: We thank you for the helpful comment. Accordingly, we calculated the rate constants for the degradation of dye solution and revised to include it in the text (lines 293-295 and 331-333 and Table 2).

Reviewer 5 Report

Peer review on manuscript

"Solar-light-driven efficient ZnO–SWCNT photocatalyst for the degradation of a persistent 3 water pollutant organic dye"

Manuscript ID: Catalyst-499196

Decision on the manuscript: Accepted in the present form.

The study carried out by the authors is properly presented. The introduction reveals a previous knowledge and research about the topic what it is translated into a proper description on the preparation of the materials. The discussion of the results makes clear the intended practical application of their research as it is a novelty, which can become into a useful application in the field of photocatalysis or improving the actual systems with similar purposes. The main analyses on the materials, specially in the comparison of ZnO-SWCNTs composites with the bare materials, that the authors have employed are appropriate in my opinion, as well as the conclusions are mostly supported by the results. In agreement with the manuscript, the Figures shows clearly the content described in the text.

I see no major flaws in the publication of this manuscript than a small revision on the nomenclature used for the XPS description. In my opinion the electronic states should be noted with a subscript (e.g. 2p1/2 instead of 2p1/2), as well as a review on the subscripts in line 52. Beside these slight modifications I consider that the manuscript is ready for its publication in the journal.

Author Response

Response to Reviewer-5’s comments

The study carried out by the authors is properly presented. The introduction reveals a previous knowledge and research about the topic what is translated into a proper description on the preparation of the materials. The discussion of the results makes clear the intended practical application of their research as it is a novelty, which can become into a useful application in the field of photocatalysis or improving actual systems with similar purposes. The main analyses on the materials, specially in the comparison of ZnO-SWCNTs composites with the bare materials, that the authors have employed are appropriate in my opinion, as well as the conclusions are mostly supported by the results. In agreement with the manuscript, the Figures show clearly the content described in the text.       

I see no major flaws in the publication 01 this manuscript than a small revision on the nomenclature used for the XPS description. In my opinion the electronic states should be noted with a subscript (e.g. 2p1/2 instead of 2p1/2) as well as a review on the subscripts in line 52. Besides these slight modifications I consider that the manuscript is ready for its publication in the journal.

Author Reply: We are thankful to you for this encouraging and helpful comment. In response, we corrected the electronic states description in the text (lines 215 and 222).

Round  2

Reviewer 2 Report

The answers from authors and the revised manuscript is acceptable at present form.

Author Response

      We are very thankful to you for your kind acceptance.

Reviewer 3 Report

I apologise if the Authors were not happy with the "form" of the first round of revisions. 

I carefully read the rebuttal letter and the new version of the manuscript. Manuscript that is not suitable for publication in Catalysts -- keep in mind that this is a journal with an eye on (photo)catalysis  . 

My comments, trying to be as much unambiguous as I can: 

Authors claim to have synthesised an efficient photocatalyst for the (photocatalytic) removal of an organic dye (methylene blue, here names as a persistent water pollutant). Let us assume that the the claim about the synthesis method is true. 

Authors then claim that "the photocatalytic performance of the constructed nanocomposite was explored through its use as a catalyst for the decomposition of MB under sunlight". 

And that: "the decomposition of MB via photocatalysis was performed under uninterrupted sunlight for 120 min within the temperature range 8–11°C (10:30 am to 12:30 pm in the outdoor environment) in  February. The transmission of UV light into the reaction vial was prevented by a UV cut-off filter such that only visible and near-infrared wavelengths could affect the reaction."

Thus, Authors used merely visible light for the photocatalytic reactions. What about the reproducibility of the experiments (experiments were done in February in the morning, as I am not mistaken)? What about the radiant flux per unit area (irradiance) reaching the photocatalyst? This information has to be given.

As well known, and as suggested in the previous round of revisions, the system organic-dye/photocatalyst is not the best system to prove a real photocatalytic reaction -- I previously suggested a paper: dye-sensytisation effect might happen. This means that a direct photo-oxidation of the ZnO-CNT might have not happened. Authors should, at least, demonstrate that a direct photo-oxidation of their material happened. This should be done using techniques such as TOC and HPLC-MS. 

On the contrary, they simply used a spectrometer to show a mere decolouration. That is not photocatalysis. At least: Authors did not proved a real photocatalytic mechanism. Other molecules (transparent) should be used instead. 

Furthermore, the apparent energy band-gap of the produced materials are not calculated (Tauc-plot, differential method, inter alia). Authors simply showed  the absorption spectra of their material strongly dispersed in ethanol. However, no visible-light absorption was proved -- only an absorption peak located at 384 nm for the ZnO-SWCNT specimen: that is no visi-light. BTW: *absorbance* is dimensionless. 

Going on. In the sentence: "The decomposition rate of MB by the pristine ZnO nanorods or pristine SWCNTs appears prolonged, the reaction being chiefly assigned to the photosensitization of the dye. By contrast, the ZnO–SWCNT nanocomposites show remarkably higher hotocatalytic performance (Figure 8a) than the pristine ZnO (Figure 8b) and the pristine SWCNTs (Figure 8c). These findings imply that compositing SWCNTs with ZnO nanorods promotes the separation of photoinduced electron–hole pairs, which is critical for enhancing the photodegradation ability of the substrate."

Why Authors are so sure that with specimen ZnO they have photosensitisation, and with ZnO-SWCNT photocatalysis? Which are the experimental proofs (see above)?

Why they do claim that "compositing SWCNTs with ZnO nanorods promotes the separation of photoinduced electron–hole pairs" ? Which is the evidence? Authors should prove it (PL spectra, transient photocurrent response experiments, for instance, all using visible-light as the light source). 

Table 1: 3 papers to compare the experiments of the Authors is a very small number. Furthermore, to have a real comparison, Authors should report also the irradiance reaching the photocatalysts. And the conditions should be the very same. All in all, that Table, as well as the Table 2 in the amended version is meaningless. 

Figure 9: Authors should report the energy gap of ZnO relative to the vacuum or to the NHE, for instance. The position of the CB and VB -- relative to some electrode or vacuum. And locate the SWCNT accordingly. Figure 9 is too simplistic as it is. 

Minor: 
1. Debye-Scherrer equation does not exist. Scherrer equation instead. However, did Authors take into account the instrumental contribution to the FWHM? Why just the (h0l) direction? Why not, for instance also the (00l)? Which the error in the measurement? Which the Scherrer constant? And why? XRD pattern, not spectra. 

2. Why only one composition (i.e. ZnO:SWCNT = 1.96:1)?

Author Response

Please see the attached response.

Reviewer 4 Report

The corrections made by the Authors have improved the quality of the manuscript, but they are not comprehensive. However, taking into account the limited research facility, and diligence of the Authors in the preparation of the response, I recommend this article to be published in the Catalysts.

Author Response

(The authors gave the same response as above.)
